# A Model cortical network for spatiotemporal sequence learning and prediction

## Abstract

In this paper we developed a hierarchical network model, called Hierarchical Prediction Network (HPNet) to understand how spatiotemporal memories might be learned and encoded in a representational hierarchy for predicting future video frames. The model is inspired by the feedforward, feedback and lateral recurrent circuits in the mammalian hierarchical visual system. It assumes that spatiotemporal memories are encoded in the recurrent connections within each level and between different levels of the hierarchy. The model contains a feed-forward path that computes and encodes spatiotemporal features of successive complexity and a feedback path that projects interpretation from a higher level to the level below. Within each level, the feed-forward path and the feedback path intersect in a recurrent gated circuit that integrates their signals as well as the circuit's internal memory states to generate a prediction of the incoming signals. The network learns by comparing the incoming signals with its prediction, updating its internal model of the world by minimizing the prediction errors at each level of the hierarchy in the style of *predictive self-supervised learning*. The network processes data in blocks of video frames rather than a frame-to-frame basis. This allows it to learn relationships among movement patterns, yielding state-of-the-art performance in long range video sequence predictions in benchmark datasets. We observed that hierarchical interaction in the network introduces sensitivity to memories of global movement patterns even in the population representation of the units in the earliest level. Finally, we provided neurophysiological evidence, showing that neurons in the early visual cortex of awake monkeys exhibit very similar sensitivity and behaviors. These findings suggest that predictive self-supervised learning might be an important principle for representational learning in the visual cortex.

## 1 Introduction

While the hippocampus is known to play a critical role in encoding episodic memories, the storage of these memories might ultimately rest in the sensory areas of the neocortex (McClelland & McNaughton, 1999). Indeed, a number of neurophysiological studies suggest that neurons throughout the hierarchical visual cortex, including those in the early visual areas such as V1 and V2, might be encoding memories of object images (Huang et al., 2018) and of visual sequences in cell assemblies (Yao et al., 2007; Han et al., 2008; Xu et al., 2012; Cooke & Bear, 2014; 2015). As specific priors, these memories, together with the generic statistical priors encoded in receptive fields and connectivity of neurons, serve as internal models of the world for predicting incoming visual experiences. In fact, learning to predict incoming visual signals has also been proposed as an objective that drives representation learning in a recurrent neural network in a self-supervised learning paradigm, where the discrepancy between the model's prediction and the incoming signals can be used to train the network using backpropagation, without the need of labeled data (Elman, 1990; Mathieu et al., 2015; Villegas et al., 2017; Srivastava et al., 2015; O'Reilly et al., 2014; Lee, 2015).

In computer vision, a number of hierarchical recurrent neural network models, notably PredNet (Lotter et al., 2016) and PredRNN++ (Wang et al., 2018), have been developed for video prediction with state-of-the-art performance. PredNet, in particular, was inspired by the neuroscience principle of predictive coding (Mumford, 1991; Rao & Ballard, 1999; Lee, 2015; Dijkstra et al., 2017; Friston, 2018). It learns a LSTM (long short-term memory) model at each level to predict the prediction

errors made in an earlier level of the hierarchical visual system. Because the error representations are sparse, the computation of PredNet is very efficient. However, the model builds a hierarchical representation to model and predict its own errors, rather than learning a hierarchy of features of successive complexities and scales to model the world. The lack of a compositional feature hierarchy hampers its ability in long range video predictions.

Here, we proposed an alternative hierarchical network architecture. The proposed model, HPNet (Hierarchical Prediction Network), contains a fast feedforward path, instantiated currently by a fast deep convolutional neural network (DCNN) that learns a representational hierarchy of features of successive complexity, and a feedback path that brings a higher order interpretation to influence the computation a level below. The two paths intersect at each level through a gated recurrent circuit to generate a hypothetical interpretation of the current state of the world and make a prediction to explain the bottom-up input. The gated recurrent circuit, currently implemented in the form of LSTM, performs this prediction by integrating top-down, bottom-up, and horizontal information. The discrepancy between this prediction and the bottom-up input at each level is called prediction error, which is fed back to influence the interpretation of the gated recurrent circuits at the same level as well as the level above.

To facilitate the learning of relationships between movement patterns, HPNet processes data in the unit of a spatiotemporal block that is composed of a sequence of video frames, rather than frame by frame, as in PredNet and PredRNN++. We used a 3D convolutional LSTM at each level of the hierarchy to process these spatiotemporal blocks of signals (Choy et al., 2016), which is a key factor underlying HPNet's better performance in long range video prediction.

In the paper, we will first demonstrate HPNet's effectiveness in predictive learning and its competency in long range video prediction. Then we will provide neurophysiological evidence showing that neurons in the early visual cortex of the primate visual system exhibit the same sensitivity to memories of global movement patterns as units in the lowest modules of HPNet. Our results suggest that predictive self-supervised learning might indeed be an important strategy for representation learning in the visual cortex, and that HPNet is a viable computational model for understanding the computation in the visual cortical circuits.

## 2 RELATED WORKS

Our objective is to develop a hierarchical cortical model for predictive learning of spatiotemporal memories that is competitive both for video prediction, and for understanding the learning principles and the computational mechanisms of the hierarchical visual system. In this regard, our model is similar conceptually to Ullman's counter-stream model (Ullman, 1995), Mumford's analysis by synthesis framework (Mumford, 1992), and Hawkin's hierarchical spatiotemporal memory model (HTM) (Hawkins & George, 2006) for hierarchical cortical processing. At a conceptual level, it can also be considered as a deep learning implementation of hierarchical Bayesian inference model of the visual cortex (Lee & Mumford, 2003; Dayan et al., 1995; Kersten & Yuille, 2003).

HPNet integrates ideas of predictive coding (Mumford, 1992; Rao & Ballard, 1999; Lotter et al., 2016) and associative coding (McClelland & Rumelhart, 1985; Grossberg, 1987). It differs from the predictive coding models (Rao & Ballard, 1999; Lotter et al., 2016) in that it learns a hierarchy of feature representations in the feedforward path to model features in the world as in normal deep convolutional neural networks (DCNN). PredNet, on the other hand, builds a hierarchy to model successive prediction errors of its own prediction of the world. PredNet is efficient because its convolution is operated on sparse prediction error codes, but we believe lacking a hierarchical feature representation limits its ability to model relationships among more global and abstract movement concepts for longer range video prediction. We believe having a fast bottom-up hierarchy of spatiotemporal features of successive scale and abstraction will allow the system to see further into the future and make better prediction.

A key difference between the genre of predictive learning models (HPNet, PredNet) and the earlier predictive coding models implemented by Kalman filters (Rao & Ballard, 1999) or associative coding models implemented by interactive activation (McClelland & Rumelhart, 1985; Grossberg, 1987) is that the synthesis of expectation is not done simply by the feedback path, via weight matrix multiplication, but by local gated recurrent circuits at each level. This key feature makes this

genre of predictive learning models more powerful and competent in solving real computer vision problems.

The idea of predictive learning, using incoming video frames as self-supervising teaching labels to train recurrent networks, can be traced back to Elman (1990). Recently, there has been active exploration of self-supervised learning in computer vision (Palm, 2012; O'Reilly et al., 2014; Goroshin et al., 2015; Srivastava et al., 2015; Patraucean et al., 2015; Vondrick et al., 2016), particularly in the area of video prediction research (Mathieu et al., 2015; Kalchbrenner et al., 2017; Tulyakov et al., 2017; Xu et al., 2018; Oh et al., 2015; Villegas et al., 2017; Lee et al., 2018; Wichers et al., 2018). The large variety of models can be roughly grouped into three categories: autoencoders, DCNN, and hierarchy of LSTMs. Some models also involve feedforward and feedback paths, where the feedback paths have been implemented by deconvolution, autoencoder networks, LSTM or adversary networks (Finn et al., 2016; Lotter et al., 2016; Wang et al., 2017; 2018). Some other models, such as variational autoencoders, allowed multiple hypotheses to be sampled (Babaeizadeh et al., 2017; Denton & Fergus, 2018).

PredRNN++ (Wang et al., 2018) is the state-of-the-art hierarchical model for video prediction. It consists of a stack of LSTM, with the LSTM at one level providing feedforward input directly to the LSTM at the next level, and ultimately predicting the next video frame at its top level. Thus, its hierarchical representation is more similar to an autoencoder, with the intermediate layers modeling the most abstract and global spatiotemporal memories of movement patterns and the subsequent layers representing the unfolding of the feedback path into a feedforward network with its top-layer's output providing the prediction of the next frame. PredRNN++ does not claim neural plausibility, but it offers state-of-the-art performance for benchmark performance evaluation, with documented comparisons to other approaches.

Recent single-unit recording experiments in the inferotemporal cortex (IT) of monkeys have shown that neurons responded significantly less to predictable sequences than to novel sequences (Meyer & Olson, 2011; Meyer et al., 2014; Ramachandran et al., 2017), suggesting that neural activities might signal prediction errors. The novel neurophysiolgical experiment we presented here demonstrated similar prediction suppression effects in the early visual cortex of monkeys for well-learned videos, suggesting neuronal sensitivity to memories of global movement patterns and scene context in the earliest visual areas. This is consistent with other recent studies that showed neurons in mouse V1 might be able to encode some forms of spatiotemporal memories in their recurrent circuits (Han et al., 2008; Xu et al., 2012; Cooke & Bear, 2015).

## 3 HIERARCHICAL PREDICTION NETWORK

### 3.1 CORTICAL MODULE

HPNet is composed of a stack of Cortical Modules (CM). Each CM can be considered as a visual area along the ventral stream of the primate visual system, such as V1, V2, V4 and IT. We used four Cortical Modules in our experiment. The network contains a feedforward path that is realized in a deep convolutional neural network (DCNN), a stack of Long Short Term Memory (LSTM) modules that link the feedforward path and the feedback path together.

Figure 1 (a) shows two CMs stacked on top of each other. The feedforward path performs convolution (indicated by $\star$) on the input spatiotemporal block $I_l$ with a kernel to produce $R_l$, where $l$ indicates CM level. $R_l$ is then down-sampled to provide the input $I_{l+1}$ for $CM_{l+1}$ for another round of convolution in the feedforward path. $I_{l+1}$ also goes into $LSTM_{l+1}$ (Lhe STM in $CMt_{l+1}$). In each $CM_l$ level, the bottom-up input $I_l$ is compared with the prediction $P_l$ generated from the interpretation output $H_l$ of $LSTM_l$. The prediction error signal is transformed by a convolution into $E_l$, which is fed back to both $LSTM_l$ and $LSTM_{l+1}$ to influence their generation of new hypotheses $H_l$ and $H_{l+1}$. To make the timing relationship between the different interacting variables more explicit, we now use $k$ to indicate time step or, equivalently, the video input frame. $LSTM_l$ at step $k$ integrates the bottom-up feature input $R_{l-1}^k$, the top-down feedback of the higher CM's LSTM's output $H_{l+1}^k$, and the prediction errors $E_{l-1}^k$ and $E_l^{k-d}$ to generate new hypothesis output $H_l^k$, which is then transformed into a new prediction $P_l^k$, where $d$ is is the number of frames in each spatiotemporal block (details in Algorithm 1).

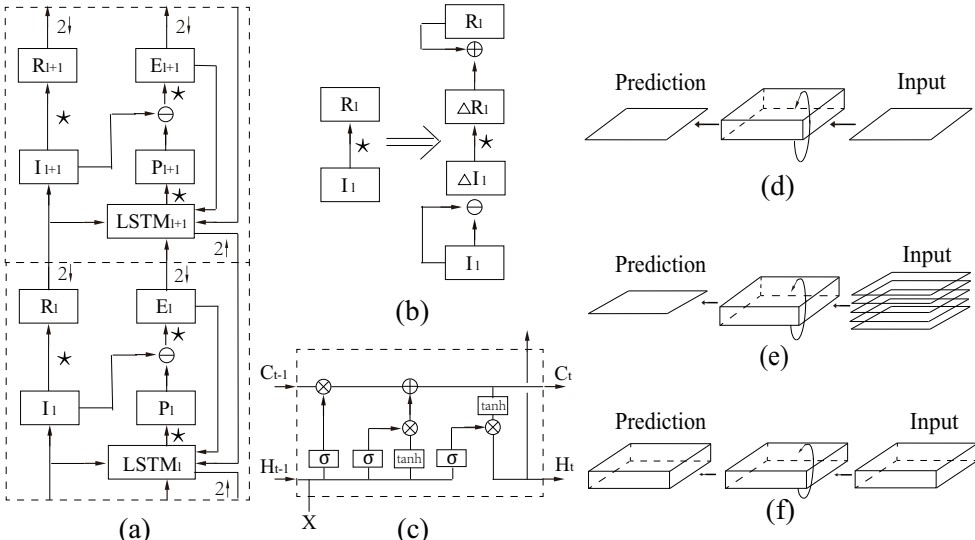

Figure 1: (a) Two Cortical Modules stacked on top of each other. The input $I_1$ would be the spatiotemporal block of video frames. The $\star$ notation means a convolution along that path. $2\uparrow$ indicates up-sampling or expansion operation. $2\downarrow$ means down-sample or reduction in resolution. $\ominus$ indicates comparator or subtraction operation; (b) The DCNN analysis path is actually implemented in a sparsified convolution scheme to speed up bottom-up processing; (c) Detailed structure of the LSTM used. $C_t$ is the internal state, and $H_t$ is the output. X is external input, which includes multiple sources in our model. (d) Frame-by-frame method; (e) Block-by-frame method; and (f) Block-by-block method, where left and right part indicates output and input with the middle indicating 2D or 3D convolution LSTM.

## 3.2 SPARSE CONVOLUTION

The feedforward DCNN path in Figure 1 (a) runs much faster if the input to each convolution layer is made sparse, as shown in Pan et al. (2018). In video processing, a scheme has been proposed by Liu et al. (2017); Dave et al. (2017); Pan et al. (2018) to sparsify the input of a convolution layer by performing convolution on the difference $\Delta I_l^k = I_l^k - I_l^{k-1}$ between two consecutive frames, where $k$ indicates the k-th frame. The resulting $\Delta R_l^k$ is added back to the representation of the last time frame $R_l^{k-1}$ to recover the representation at the current frame $R_l^k$. This allows the network to maintain a full higher order representation $R$ at all times in the next layer while enjoying the benefit of fast computation on sparse input. In their scheme (Pan et al., 2018), the first frame $I^{k=0}$ was convolved with a set of dense convolution kernels and then the subsequent frames were convolved with a set of sparse convolution kernels. For parsimony and neural plausibility, we used the same set of sparse kernels for processing both the first full frame and the subsequent temporal-difference frames, at the expense of incurring some inaccuracy in our prediction of the first few frames.

## 3.3 SPATIOTEMPORAL BLOCKS AND 3D CONVOLUTION

The input data of our network model is a sequence of video frames or a spatiotemporal block. For our implementation, each block contains 5 video frames. If we consider that each frame corresponds roughly to 25 ms, this would translate into 125 ms, in the range of the length of temporal kernel of a cortical neuron. Our convolution kernel is in three dimension, processing the video by spatiotemporal blocks. The block could slide in time with a temporal stride of one frame or a stride as large as the length of the block $d$. The LSTM is a 3D convolutional LSTM (Choy et al., 2016) because of 3D convolution and spatiotemporal blocks. Convolution LSTM (Shi et al., 2015), in which Hadamard product in LSTM is replaced by a convolution, has greatly improved the performance of LSTM in many applications. Earlier video prediction models (e.g. PredNet, PredRNN) processed video sequences frame by frame, as shown in Figure 1 (d). We experimented with different data units and approaches. In the Frame-to-Frame (F-F) approach, an input frame is used to generate one

predicted future frame (Figure 1 (d)). In the Block-to-Frame (B-F) approach (Figure 1 (e)), a block of input frames is used to generate one predicted future frame. This approach is time consuming, but provides more accurate near-range predictions. For longer-range predictions, we found using a spatiotemporal block to predict a spatiotemporal block, i.e. the Block-to-Block (B-B) approach ( Figure 1(f)), to be the most effective, because the LSTM learns the relationship between movement segments in the sequences. The details of our algorithm of the 3D convolutional LSTM is specified in Appendix A.

## 3.4 TRAINING AND LOSS FUNCTION

The entire network is trained by minimizing a loss function which is the weighted sum of all the prediction errors, with the following algorithm,

$$I_l^k = \begin{cases} MaxPool(ReLU(R_{l-1}^k)) & l > 1 \\ x_t & l = 1 \end{cases} \quad P_l^k = \begin{cases} ReLU(conv(H_l^k)) & l > 1 \\ SATLU(ReLU(conv(H_l^k))) & l = 1 \end{cases} \quad (1)$$

$$\Delta I_l^k = I_l^k - I_l^{k-d}, \ \ \Delta E_l^k = I_l^k - P_l^k \tag{2}$$

$$\Delta R_l^k = spconv(\Delta I_l^k), \ \ E_l^k = spconv(\Delta E_l^k), \ \ R_l^k = R_l^{k-d} + \Delta R_l^k \tag{3}$$

$$H_l^k = 3DconvLSTM(H_l^{k-d}, E_l^{k-d}, MaxPool(ReLU(R_{l-1}^k, E_{l-1}^k)), upsample(H_{l+1}^k)) \tag{4}$$

$$L_{loss} = \sum_k \lambda_k \sum_l \frac{\lambda_l}{n_l} \sum_{n_l} \Delta E_l^k \tag{5}$$

where $x_t$ is the input sequence, $H_l^k$ is the output of LSTM, $P_l^k$ is the prediction, SATLU is a saturating non-linearity set at the maximum pixel value: $SATLU(x; p_{max}) := \min(p_{max}, x)$, $spconv$ is sparse convolution, $\lambda_k$ and $\lambda_l$ are weighting factors by time and CM level, respectively, and $n_l$ is the number of units in the $l$th CM level, and $d$ is the number of frames in each spatiotemporal block. The full algorithm is shown in Algorithm 1.

## 4 EXPERIMENTAL RESULTS

In this section, we first evaluate the performance of our model in video prediction using two benchmark datasets: (1) synthetic sequences of the Moving-MNIST database and (2) the KTH[1] real world human movement database. We then investigate the representations in the model to understand how recurrent network structures have impacted on the feedforward representation. We finally compare the temporal activities of neurons in the network model with that of neurons in the visual cortex of monkeys, in video sequence learning, to evaluate the plausibility of HPNet.

Since for video prediction, PredNet is the most neurally plausible model and PredRNN++ provides state-of-the-art computer vision performance, we will compare HPNet's performance with these two network models. Because these two models work on frame-to-frame basis, we implemented three versions of our network for comparison: (1) Frame-to-Frame (F-F), where we set our data spatiotemporal block size to one frame and used 2D convLSTM instead of 3D convLSTM to predict the next frame based on the current frame; (2) Block-to-Frame (B-F), where we used a sliding block window to predict the next frame based on the current block of frames; (3) Block-to-Block (B-B), where the next spatiotemporal block was predicted from the current spatiotemporal block (Figure 1 (d)).

We trained all five networks using 40-frame sequences extracted from the two databases in the same way as described in (Lotter et al., 2016; Wang et al., 2018). We then compared their performance in predicting the next 20 frames when only the first 20 frames were given. The test sequences were drawn from the same dataset but not in the training set. The common practice in PreNet and PredRNN++ for predicting future frames when input is no longer available is to make the prediction of the last time step the next input and use that to generate prediction of the next time step. All models tested have four modules (layers). All three versions of our model and PredNet used the same number of feature channels in each layer, optimized by grid search, i.e. (16,32,64,128) for the Moving-MNIST dataset, and (24,48,96,192) for the KTH dataset. For PredRNN++, we used

---

[1] http://www.nada.kth.se/cvap/actions/

---

**Algorithm 1** The algorithm of our model

---

**Input:** $I_1^k \leftarrow x_t$
1: **for** $t = 1$ to $T$ **do**
2:     **for** $l = L$ to $1$ **do**                                             ▷ Top-down procedure
3:         **if** $l = L$ **then**
4:             $H_l^k = 3DconvLSTM(H_l^{k-d}, E_l^{k-d}, MaxPool(ReLU(R_{l-1}^k, E_{l-1}^k)))$
5:         **else**
6:             $H_l^k = 3DconvLSTM(H_l^{k-d}, E_l^{k-d}, MaxPool(ReLU(R_{l-1}^k, E_{l-1}^k)), upsample(H_{l+1}^k))$
7:         **end if**
8:     **end for**
9:     **for** $l = 1$ to $L$ **do**                                              ▷ Bottom-up procedure
10:         **if** $l = 1$ **then**
11:             $I_l^k = x_t, \ P_l^k = SATLU(ReLU(conv(H_l^k)))$
12:         **else**
13:             $I_l^k = MaxPool(ReLU(R_{l-1}^k)), \ P_l^k = ReLU(conv(H_l^k))$
14:         **end if**
15:         $\Delta I_l^k = I_l^k - I_l^{k-d}, \Delta R_l^k = spconv(\Delta I_l^k), R_l^k = R_l^{k-d} + \Delta R_l^k$
16:         $\Delta E_l^k = I_l^k - P_l^k, E_l^k = spconv(\Delta E_l^k)$
17:     **end for**
18: **end for**

---

the same architecture and feature channel numbers provided by Wang et al. (2018). All kernel sizes are either $3 \times 3$ (for F-F) or $3 \times 3 \times 3$ (for B-F and B-B) for all five models. The input image frame's spatial resolution is $64 \times 64$.

The models were trained and tested on GeForce GTX TITAN X GPUs. We evaluated the prediction performance based on two quantitative metrics: Mean-Squared Error (MSE) and the standard Structural Similarity Index Measure (SSIM) (Wang et al., 2004) of the last 20 frames between the predicted frames and the actual frames. The values of SSIM range from -1 to 1, with a larger value indicating greater similarity between the predicted frames and the actual future frames.

### 4.1 SYNTHETIC SEQUENCE PREDICTION ON THE MOVING-MNIST DATASET

We randomly chose subsets of digits in the Moving MNIST[2] dataset in which the video sequences contain two handwritten digits bouncing inside a frame of $64 \times 64$ pixels. We extracted 40-frame sequences at random starting frame position in the video in the same way as in Srivastava et al. (2015) (followed by PredNet and PredRNN++). This extraction process is repeated 15000 times, resulting in a training set of 10000 sequences, a validation set of 2000 sequences, and a testing set of 3000 sequences.

Figure 2 and Table 1 compare the results of different models on the Moving-MNIST dataset. There are 40 frames in total and we show the results every two frames. The yellow vertical line in the middle represents the border between the first 20 and the last 20 predicted frames by various models. We can see B-F achieves better performance than B-B in short term prediction task when actual input frames are provided, but B-B outperforms B-F in the longer range prediction, reflecting learning of the relationships at the movement levels by the 3D convLSTM. B-F doing better than F-F confirmed that the spatiotemporal block data structure provides additional information for modeling movement tendency. Finally, we found that even F-F achieved better prediction results than PredNet, suggesting that a feature hierarchy might be more useful than a hierarchy of predicted errors. Finally, our B-B network outperformed the state-of-the-art PredRNN++.

### 4.2 REAL-WORLD SEQUENCE PREDICTION ON THE KTH DATASET

Schüldt et al. (2004) introduced the KTH video database which contains 2391 sequences of six human actions: walking, jogging, running, boxing, hand waving, and hand clapping, performed by 25 subjects in four different scenarios. We divided video clips across all 6 action categories into a training set of 108717 sequences (persons #1-16) and a test set of 4086 sequences (persons #17-

---

[2] http://yann.lecun.com/exdb/mnist/

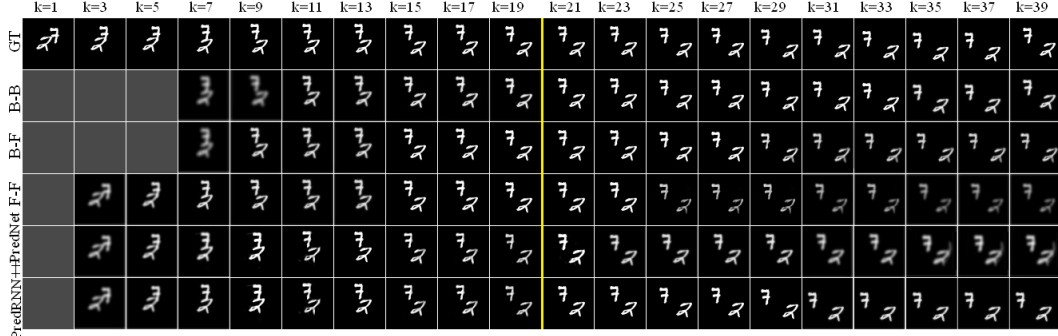

Figure 2: Video prediction results on Moving-MNIST dataset, where the first row to last row are ground truth (GT), results from three different version of HPNet (block-to-block (B-B), block-to-frame (B-F), frame-to-frame (F-F)), PredNet, and PredRNN++, respectively. k=1 to k=19 are predicted frames of the models when the input frames were available. k=21 to k=39 are the "dead-reckoning" predicted frames of the model when there are no input.

Table 1: Comparison Results of different methods on Moving-MNIST datatset for long time prediction experiment.

| Method | SSIM | MSE |
|---|---|---|
| Ours(B-B) | **0.915** | **65.2** |
| Ours(B-F) | 0.793 | 73.2 |
| CM+ConvLSTM (F-F) | 0.692 | 89.5 |
| PredNet (Lotter et al., 2016) | 0.658 | 101.2 |
| PredRNN++ (Wang et al., 2018) | 0.872 | 69.4 |

Table 2: Comparison Results of different methods on the KTH datatset for long time prediction experiment.

| Method | SSIM | MSE |
|---|---|---|
| Ours(B-B) | **0.882** | **80.3** |
| Ours(B-F) | 0.784 | 93.1 |
| CM+ConvLSTM (F-F) | 0.701 | 103.4 |
| PredNet (Lotter et al., 2016) | 0.656 | 108.9 |
| PredRNN++ (Wang et al., 2018) | 0.865 | 86.7 |

25) as was done in Wang et al. (2018), except we extracted 40-frame sequences. We center-cropped each frame to a $120 \times 120$ square and then re-sized it to input frame size of $64 \times 64$.

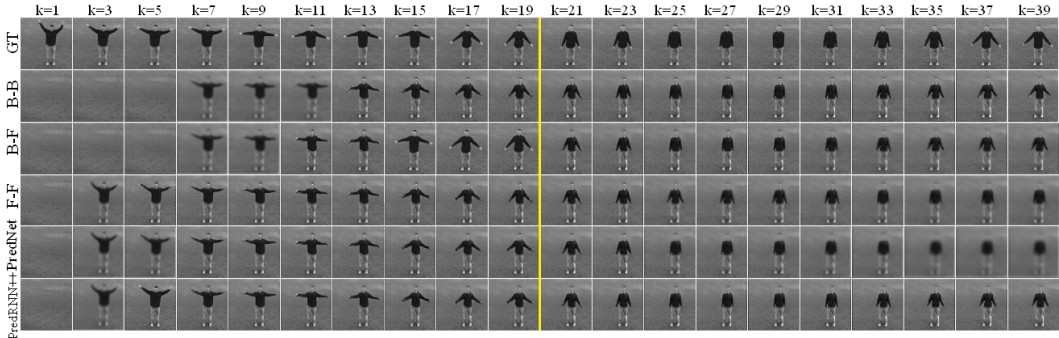

Figure 3: Video prediction results on the KTH dataset, where the first row to last row are ground truth (GT), results from block-to-block (B-B), block-to-frame (B-F), frame-to-frame (F-F), PredNet, and PredRNN++, respectively, same format as Figure 2.

Figure 3 and Table 2 compared the results of the different models on the KTH dataset, essentially reproducing all the observations we made based on the Moving-MINST dataset (Figure 2). B-B outperformed all tested models in the long range video prediction task. Figure 4 (a) and (b) compared the video prediction performance of the different models in terms of the "dead-reckoning frames" to be predicted when only the first twenty frames were provided for the two datasets. The results show that, in both cases, B-B is far more effective than B-F in long range video prediction. Figure 4 (c) showed that the ratio of SSIM and training time peaks at a 4-module network. The SSIM of a 5-module network was about the same as that of a 4-module network but took longer time to converge. The B-F, with a sliding window of a single frame stride, took much longer to train yet still

under-performed. Figure 4 (d) showed SSIM performance and training time of the different models. It shows that the B-B (sparse) version of HPNet took only 10% longer to train than PredRNN++ even though it has more loops into the networks and has to process spatiotemporal blocks. Both PredRNN++ and HPNet require twice amount of the training time relative to PredNet, illustrating the computational efficiency of using sparse codes. Sparsifying our DCNN feedforward path reduced our B-B network's training time by 13% (comparing B-B (sparse) versus B-B (non-sparse) in Figure 4 (d)).

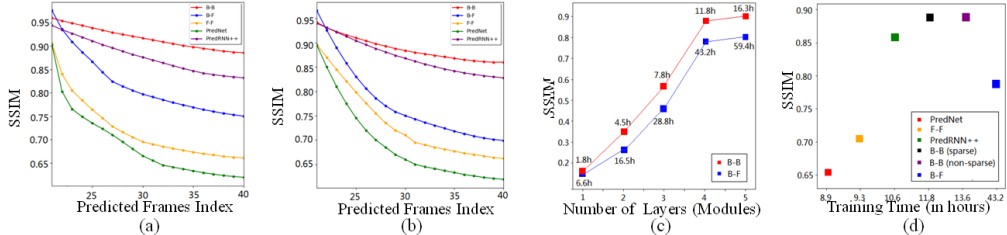

Figure 4: (a) Comparison of the prediction results of the five models for the Moving-MINST dataset on the last 20 frames in structural similarity measures (SSIM). (b) Comparison of the prediction results on the KTH datset. (c) Comparison of the performance (and training time) of the B-B and the B-F networks as a function of the number of modules in the network. (d) Training time versus SSIM performance of the different models. Note, the training time (x) axis not in a linear scale.

To understand the importance of the hierarchical representation and recurrent feedback in the model, we trained the B-B network with different numbers of modules and then used t-SNE (van der Maaten & Hinton, 2008) to visualize the representation $R$ in the different modules of the various networks in response to the last of the 20 future dead-reckoning frames of 600 testing sequences belonging to the six movements in the KTH dataset. The results are shown in Figure 5. We observed that having more higher modules introduced cluster of global movement patterns in the representation units even in the earliest module (Figure 5 (a) versus Figure 5 (e)), which resulted in significant decoding accuracy improvement in the classification of the six classes of movement patterns, from chance (16%) to 26%, based on the unit activities in the first module alone. The representations of the top module of the 4-module network provide a decoding accuracy of 63%, suggesting that the HPNet has learned semantically meaningful hierarchical spatiotemporal feature representations (see Appendix B for details) and can learn movement-to-movement relationships for making better long range video predictions (see also Kheradpisheh et al. (2018)). Decoding results indicate that higher order semantic representations of the global movement patterns are significantly weaker or absent in the hierarchical representations of PredRNN or PredNet respectively (see Appendix B for details).

## 4.3 VISUAL SEQUENCE LEARNING EFFECTS IN THE VISUAL CORTEX

Hierarchical feedback in HPNet endows the representations in the earliest Cortical Modules with sensitivity to global movement and image patterns, despite these units' very localized receptive fields, particularly R in the feedforward path (Figure 5). Could the neurons in the early visual areas of the mammalian hierarchical visual systems behave in a similar way, becoming sensitive to the memory of global movement patterns of familiar movies?

We found this to be the case in a series of neurophysiological experiments that we have performed to study the effect of unsupervised learning of video sequences on the early visual cortical representations. Two monkeys, implanted with Gray-Matter semi-chronic multielectrode arrays (SC32 and SC96) over the V1 operculum with access to neurons in V1 and V2, participated in the experiment. Each experiment lasted for at least seven daily recording sessions. In each recording session, the monkey was required to fixate on a red dot on the screen for a water reward while a set of 40 video clips of natural scenes with global movement patterns was presented. One clip was presented per trial. Each clip lasted for 800 ms. A total of 40 clips were presented once each in a random inter-leaved fashion in a block of trials, and each block was repeated 20-25 times each day[3]. Among the

---

[3] All experimental procedures were approved by TBA University's Institutional Animal Care and Use Committee, in compliance with the guidelines set forth in the United States Public Health Service Guide for the Care and Use of Laboratory Animals.

40 movie clips tested every day, twenty of these were the same each day, designated as "Predicted set". Twenty of them were different each day, designated as "Unpredicted set". Each set consisted of 20 movies.

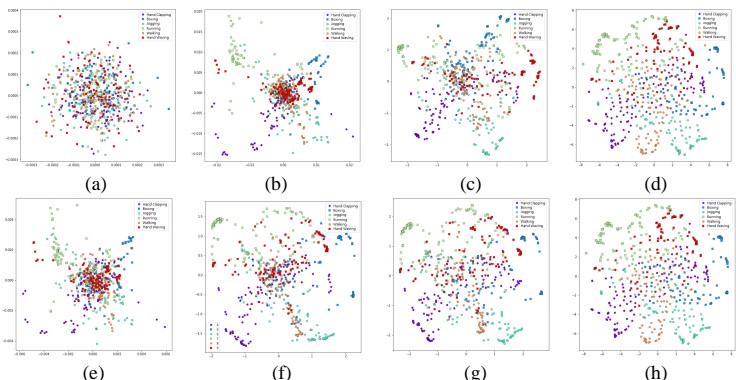

Figure 5: (a)-(d) are the top CM's R representation of networks with different number of modules, from one to four; (e)-(h) are the representation of each modules in a four-modules network, from the first module to the fourth, left to right. Better clustering leads to better decoding results of the different movement classes. Full details are in Appendix B.

The rationale for the experimental design is as follows. Given that we were recording from 30+ neurons in each session, even though the neurons have different stimulus preferences in their local receptive fields, each neuron would experience about 400 movie frames for the Predicted movie set, as well as for each of the Unpredicted movie sets. When we averaged the temporal responses of all the neurons to each of the 20-movie sets, they should be roughly the same. In the first two days of the experiment, the clips in the Predicted set were still unpredicted, hence there should have been no difference between the population averaged responses to the Predicted set and the Unpredicted set. This was indeed the case as shown in Figure 6b (top row) which compared the averaged temporal responses of the neurons to the Predicted set and to the Unpredicted set for the first two days of training in one experiment.

Interestingly, we found that after only three days of unsupervised training, with 20-25 exposures of each familiar movie per day, the neurons started to respond significantly less to predicted movies than to novel movies in the later part of their responses, starting around 100 ms post-stimulus onset, as shown in Figure 6(b) (bottom row). The evolution of daily mean of all neurons' familiarity suppression index over days is shown as the magenta curve. As the neurons became more and more familiar with the Predicted set, the prediction suppression effect gradually increased and saturated at around the sixth and seventh days. We repeated the experiments six times in two monkeys and obtained fairly consistent results. Note that the movie clips were shown in a $8^o$ aperture during the experiment. Given that the V1 and V2 neurons being studied have very local and small receptive fields ($0.5^o$ to $2^o$), it is rather improbable that the neurons would have remembered or adapted to the local movement patterns of the Predicted set within their receptive fields, as they would be experiencing millions of such local spatiotemporal patterns in their daily experience. Indeed, when the video clips were shown to the neurons through a smaller $3^o$ diameter aperture, the prediction suppression effects were much attenuated, suggesting that the neurons had indeed *became sensitive to the global context of movement patterns!*

To check whether neurons in our network behave in the same way, we performed a similar experiment on our network, pretrained with the KTH dataset. We randomly extracted 20 sequences from the BAIR dataset (Ebert et al., 2017), resized the sequence length to 40 frames and each frame size to 64×64. We separated the 20 video sequences into two sets – the Predicted set and the Unpredicted set. We averaged the responses to the two movie sets respectively of each type of neurons in the network ($E$ (prediction error units), $P$ (prediction units), and $R$ (representation units)) in each CM within the center 8×8 hypercolumns. Before training, the responses of each type of neurons are indeed the same for both movie sets (not shown, but similar to Figure 6(b) data). Then, we trained the network with the Predicted set for 2000 epochs. After training, all three types of units in each

CM exhibited the prediction suppression effect as shown in Figure 6 (c)-(h) (full details in Appendix C).

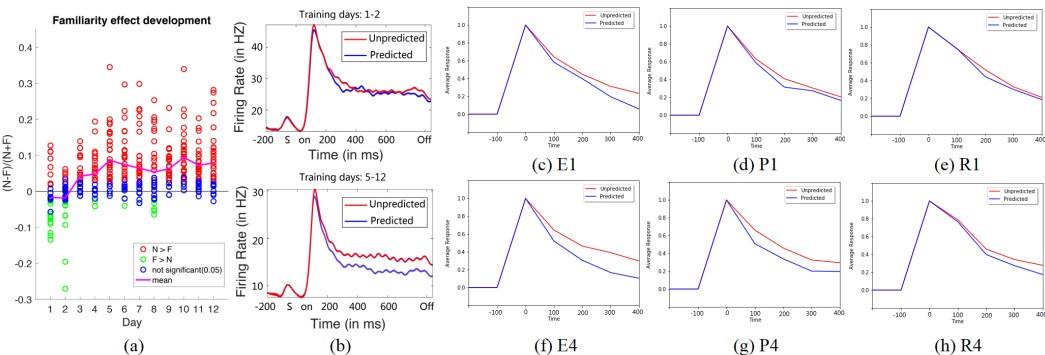

Figure 6: (a) The development of the prediction suppression effect across days in one experiment. Each dot is the prediction suppression index of a neuron. Color indicates whether the effect was significant or not (red - significant, blue - insignificant, green - significant in the opposite way) based on t-test with $p < 0.05$ as statistical significance threshold. (b) Averaged temporal responses of the V1 and V2 neurons (combined) of one monkey to Predicted set and the Unpredicted sets in the first two days (top row), showing no difference. The averaged responses (combining data from day 5 to day 12) to the Predicted set was significantly weaker than the responses to the Unpredicted sets, indicating prediction suppression. (c)-(e) Module 1's normalized averaged population responses of the three types of units to the Predicted set and the Unpredicted set. (f)-(h) Module 4's normalized averaged population responses of the three types of units.

We observed the prediction suppression effect in all three types of neurons in all the modules in the hierarchy, with the higher modules showing a stronger effect. It is not surprising that the prediction error neurons $E$ would decrease their responses as the network learns to predict the familiar movies better. It is rather interesting to find the representation neurons $R$ and the prediction neurons $P$ also exhibit prediction suppression, even though these neurons represent features rather than prediction errors. The precise reasons remain to be determined, but the fact that all neuron types in the *model* exhibited the prediction suppression effect might explain why the prediction suppression effects were commonly observed in most of the randomly sampled neurons in the visual cortex (see Figure 6a). These findings suggest that (1) predictive self-supervised learning might indeed be an important principle and mechanism by which the visual cortex learns its representations, and (2) the neurophysiological observations on prediction suppression in IT (see Appendix D) and now in the early visual cortex might be explained by this class of hierarchical cortical models.

## 5  CONCLUSION

In this paper, we developed a hierarchical prediction network model (HPNet), with a fast DCNN feedforward path, a feedback path and local recurrent LSTM circuits for modeling the counter-stream / analysis-by-synthesis architecture of the mammalian hierarchical visual systems. HPNet utilizes predictive self-supervised learning as in PredNet and PredRNN++, but integrates additional neural constraints or theoretical neuroscience ideas on spatiotemporal processing, counter-stream architecture, feature hierarchy, prediction evaluation and sparse convolution into a new model that delivers the state-of-the-art performance in long range video prediction. Most importantly, we found that the hierarchical interaction in HPNet introduces sensitivity to global movement patterns in the representational units of the earliest module in the network and that real cortical neurons in the early visual cortex of awake monkeys exhibit very similar sensitivity to memories of global movement patterns, despite their very local receptive fields. These findings support predictive self-supervised learning as an important principle for representation learning in the visual cortex and suggest that HPNet might be a viable computational model for understanding the cortical circuits in the hierarchical visual system at the functional level. Further evaluations are needed to determine definitively whether PredNet or HPNet is a better fit to the biological reality.

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

APPENDIX

## A    3D CONVOLUTIONAL LSTM

Because our data are in the unit of spatitemporal block, we have to use a 3D form of the 2D convolutional LSTM. 3D convolutional LSTM has been used by Choy et al. (2016) in the stereo setting. The dimensions of the input video or the various representations ($I$, $E$ and $H$) in any module are $c \times d \times h \times w$, where $c$ is the number of channels, $d$ is the number of adjacent frames, $h$ and $w$ specify the spatial dimensions of the frame. The 3D spatiotemporal convolution kernel is $m \times k \times k$ in size, where $m$ is kernel temporal depth and $k$ is kernel spatial size. The spatial stride of the convolution is 1. The size of the output with $n$ kernels is $n \times d \times h \times w$. We define the inputs as $X_1, ..., X_t$, the cell states as $C_1, ..., C_t$, the outputs as $H_1, ..., H_t$, and the gates as $i_t, f_t, o_t$. Our 3D convolutional LSTM is specified by the equations below, where the function of 3D convolution is indicated by $\star$ and the Hadamard product is indicated by $\circ$.

$$
\begin{aligned}
i_t &= \sigma(W_{xi} \star X_t + W_{hi} \star H_{t-1} + W_{ci} \circ C_{t-1} + b_i) \\
f_t &= \sigma(W_{xf} \star X_t + W_{hf} \star H_{t-1} + W_{cf} \circ C_{t-1} + b_f) \\
C_t &= f_t \circ C_{t-1} + i_t \circ tanh(W_{xc} \star X_t + W_{hc} \star H_{t-1} + b_c) \\
o_t &= \sigma(W_{xo} \star X_t + W_{ho} \star H_{t-1} + W_{co} \circ C_t + b_o) \\
H_t &= o_t \circ tanh(C_t)
\end{aligned}
\tag{6}
$$

## B    SEMANTIC CLUSTERING IN THE HIERARCHICAL REPRESENTATIONS

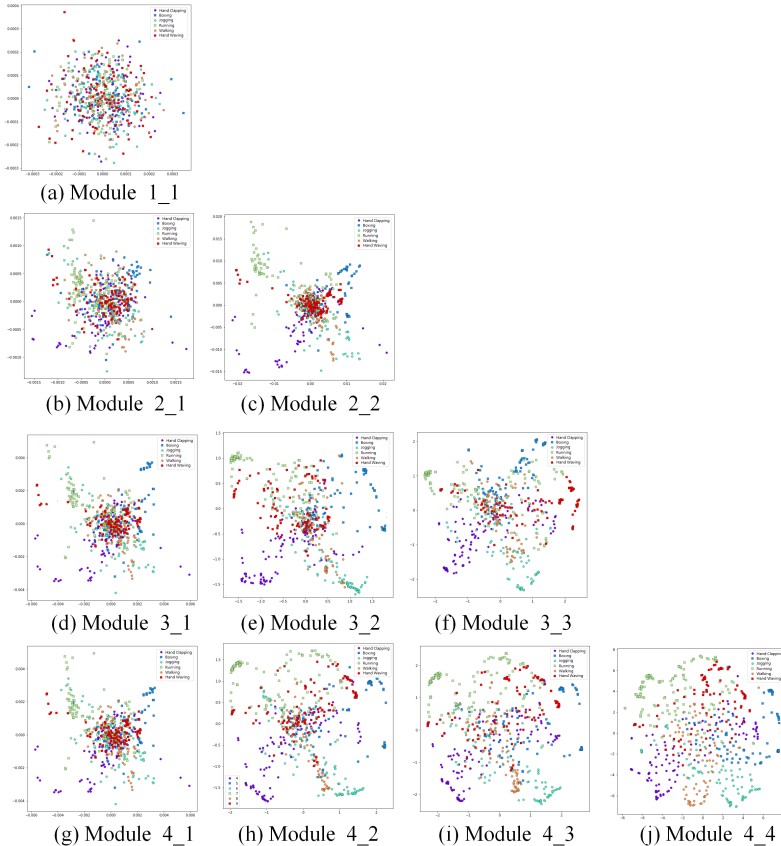

(a) Module 1_1

(b) Module 2_1    (c) Module 2_2

(d) Module 3_1    (e) Module 3_2    (f) Module 3_3

(g) Module 4_1    (h) Module 4_2    (i) Module 4_3    (j) Module 4_4

Figure 7: Visualization of R representational units of the different modules in (a) a one-module network; (b)-(c) a two-module network; (d)-(f) a three-module network; and (g)-(j) a four-module network.

Figure 7 compares the t-SNE (van der Maaten & Hinton, 2008) projection of the responses of the R representation units in the center $8 \times 8$ "hypercolumns" of the different modules for networks of different number of modules to the 6 movement classes in the KTH dataset. Partial results are shown in Figure 5 of the main text of the paper. The figures demonstrate that as more higher order modules are stacked up in the hierarchy, the semantic clustering into the six movement classes become more pronounced even in the early modules, suggesting that the hierarchical interaction has steered the feature representation into semantic clusters even in the early modules. Module 4-1 means representation of module 1 in a 4-module network.

We use linear decoding (multi-class SVM) to assess the distinctiveness of the semantiuc clusters in the representation of the different modules in the different networks. The decoding results in Table 3 shows that the decoding accuracy based on the reprsentation of module 1 has improved from chance (16%) to 26%, an improvement of 60% between a 1-module HPNet and a 4-module HPNet, and that the representation of module 4 of a 4-module HPNet can achieve a 63% accuracy in classifying the six movement classes, suggesting that the network only needs to learn to predict unlabelled video sequences, and it automatically learns reasonable semantic representations for recognition.

Table 3: Our model's decoding results of six movement classes in the KTH dataset based on R representations in different modules of networks of different number of modules. Module 4-2 means Module 2 of a 4-module HPNet.

| Representation in | Module 1-1 | Module 2-1 | Module 3-1 | Module 4-1 |
|---|---|---|---|---|
| Mean decoding accuracy | 0.16 | 0.19 | 0.21 | 0.26 |
| Representation in | Module 4-1 | Module 4-2 | Module 4-3 | Module 4-4 |
| Mean decoding accuracy | 0.26 | 0.45 | 0.57 | 0.63 |

For comparison, we also performed decoding on the output representations of each LSTM layer in the PredRNN++ and PredNet to study their representations of the six movement patterns. The results shown below indicate that the semantic clustering of the six movements is not very strong in the PredRNN++ hierarchy. We realized that this might be because the PredRNN++ behaves essentially like an autoencoder. The four-layer network effectively only has two layers of feature abstraction, with layer 2 being the most semantic in the hierarchy and layers 3 and 4 representing the unfolding of the feedback path. Decoding results indicate that the hierarchical representation based on the output of the LSTM at every layer in PredNet, which serve to predict errors of prediction errors of the previous layer, does not contain semantic information about the global movement patterns.

Table 4: PredRNN++'s decoding results of six movement classes in the KTH dataset based on representations in the different layers of the network.

| Representation in | Layer 1 | Layer 2 | Layer 3 | Layer 4 |
|---|---|---|---|---|
| Mean decoding accuracy | 0.18 | 0.23 | 0.18 | 0.16 |

Table 5: PredNet's decoding results of six movement classes in the KTH dataset based on LSTM representations in the different layers of the network.

| Representation in | Layer 1 | Layer 2 | Layer 3 | Layer 4 |
|---|---|---|---|---|
| Mean decoding accuracy | 0.16 | 0.11 | 0.10 | 0.10 |

## C    PREDICTION SUPPRESSION EFFECTS IN VIDEO SEQUENCE LEARNING IN HPNET

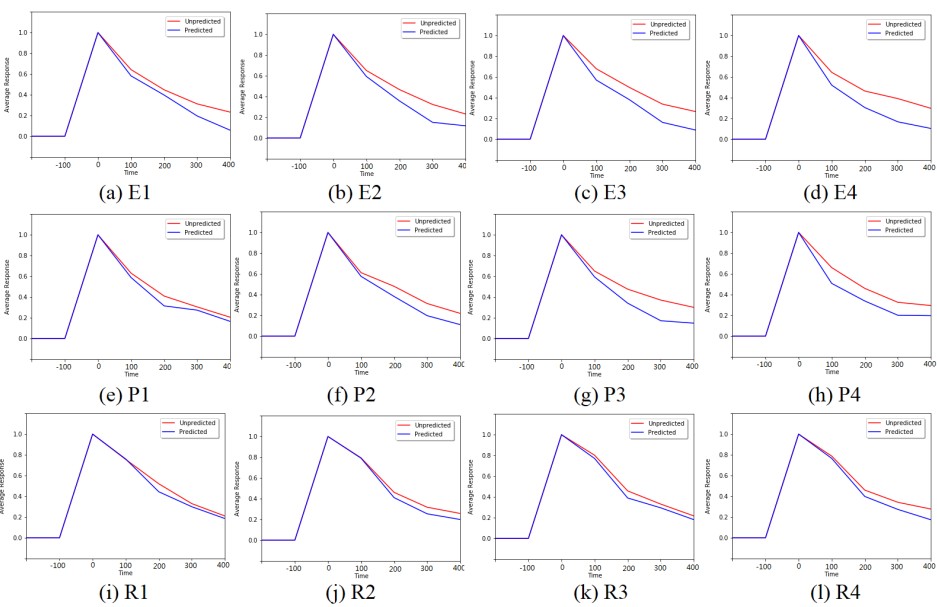

Figure 8: Results of video sequence learning experiments showing prediction suppression can be observed in $E$, $P$, and $R$ units in every module along the hierarchical network. The abscissa is time after stimulus onset - where we set each video frame to be 25 ms for comparison with neural data. The ordinate is the normalized averaged temporal response of all the units within the center 8×8 hypercolumns, averaged across all neurons and across the 20 movies in the Predicted set (blue) and the Unpredicted set (red) respectively. Prediction suppression can be observed in all types of units, though more pronounced in the E and P units.

## D    PREDICTION SUPPRESSION EFFECT IN IT NEURONS AND HPNET

HPNet readily reproduces the prediction suppression effects observed in IT neurons. Meyer & Olson (2011) trained monkeys to image pairs in a fixed order for over 800 trials for each 8 pair images, and then compared the responses of the neurons to these images in the trained order against the responses of the neurons to the same images but in novel pairings. Figure 9 shows the mean responses of 81 IT neurons during testing stage for predicted pairs and unpredicted pairs. All the stimuli are presented in both pairs. They found that neural responses to the expected second images in a familiar sequence order is much weaker than the neural responses to the image in an unfamiliar or unexpected sequence order. To evaluate whether HPNet can produce the same effect, we performed exactly the same experiments with 2000 epochs of training on the image pairs, with a gap of 2 frames, and our model produced the same results, with lower responses for the predicted second stimulus relative to the unpredicted second stimulus. Each stimulus sequence was presented first with 5 gray frames, followed by 10 frames of the first image in the pair, then 2 gray frames as gap, then 10 frames of the second image in the pair. The responses of the units to the trained set and the untrained set are the same prior to training. After training, the images when arranged in the trained order responded much less after the initial responses than the same images but arranged in unpredicted pairs. The result shown in Figure 10 duplicated the observations in Meyer & Olson (2011), the average neural response of $E$ unit is lower than the unpredicted pairs. All three types of units of NPNet exhibit prediction suppression though the effect is much weaker for the R units (see Figure 11. Lotter et al. (2018) also tested the prediction suppression effect, but their model couldn't allow any gap between the stimuli as in the experiment. Our model can handle gap because of our model is processing information in spatiotemporal blocks.

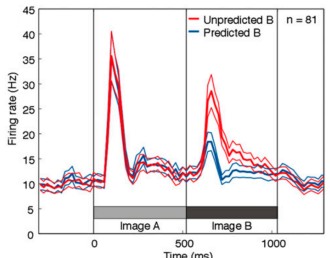

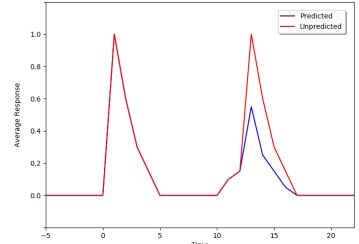

Figure 9: Prediction suppression in IT neurons ((Meyer & Olson, 2011)).

Figure 10: Prediction suppression results on $E4$ units in HPNet.

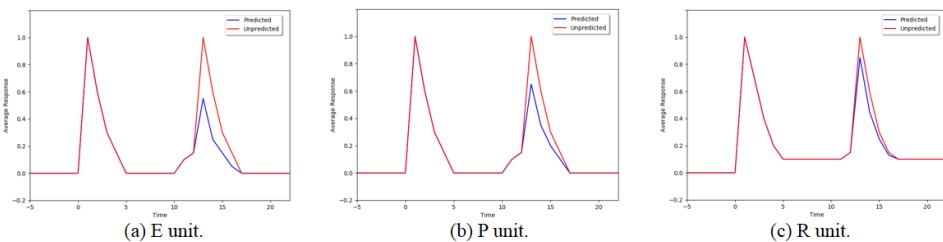

(a) E unit.

(b) P unit.

(c) R unit.

Figure 11: Prediction suppression behaviors in the E, P, and R units of module 4 of HPNet, respectively.

