# OpenReview forum: "A  Model Cortical Network for Spatiotemporal Sequence Learning and Prediction"
_ICLR.cc/2019/Conference_

### Official Review · AnonReviewer1 · 2018-10-29
**Clarity and quantification need improvement**

**Rating:** 3
**Confidence:** 3

**Review:**

This paper proposes a network architecture inspired by the primate visual cortex. The architecture includes feedforward, feedback, and local recurrent connections, which together implement a predictive coding scheme. Some versions of the network are shown to outperform the similar PredNet and PredRNN architectures on two video prediction tasks: moving MNIST and KTH human actions. Finally, the authors provide neural data from monkeys and argue that their network shows similarities to the biological data.

The paper contains intriguing ideas about the benefits of sparse and predictive coding, and the direct comparison to biological data potentially broadens the impact of the work. However, major claims are unsubstantiated, and accuracy and clarity need to be improved to make the manuscript acceptable.

Major concerns:
1. The authors claim that their architecture is more efficient because it uses sparse coding of residuals. Implementation details and some quantitative arguments, ideally benchmarks, need to be provided to show that their architecture is actually more efficient than PredRNN++ and PredNet.

2. It is unclear whether the PredRNN++ should be compared to the C-C or C-F version of the network. Does the PredRNN++ have access to as many current and future frames as the C-C net? Is this a fair comparison? Please provide a clearer description of the different versions of your network and how they relate to the baseline models. That section in particular has many confusing typos (frame-by-chunk, chunk-by-frame abbreviations mixed up).

3. In Figure 6, the authors claim that more layers lead to “better” representations. What does “better” mean? It is implied that the networks with more layers actually make the different motions more discriminable. Please quantify this. For example, a linear classifier could be trained on the neural activations. Also, how is this related to the rest of the paper? Do the authors claim that this result is unique to the proposed architecture? In that case, please provide a quantitative comparison to the PredNet or PredRNN++.

4. In Figure 9, the presentation is highly confusing. Plots (c) to (h) are clearly made to look like the monkey data in (b) (nonlinear x-axes?), but show totally different timescales (training epochs vs. milliseconds). Please explain why it makes sense to compare these timescales. Also, what does it mean for a training epoch to have a negative value?

Minor comments:
1. I don’t understand the “tension” between hierarchical feature representations and residual representations brought up in Section 2. Do the PredNet and PredRNN++ not contain a hierarchy of representations?

2. Figure 1 is not fully annotated and could be clearer. What does the asterisk mean? Why are there multiple arrows between the P’s? What do the small arrows next to the big arrows mean? Please expand the legend. Consider using colors to differentiate between components.

3. I don’t understand Figure 4c. According to the text, this plot shows “effectiveness as a function of time”, but the x-axis is labeled “Layer Number”. What does “effectiveness over time” mean? What does the y-label mean (SSIM per day?)? What is “trunk prediction” (not mentioned anywhere in the text)?

4. For Figure 9, it is pointed out that activity is expected to be lower for E neurons, but is also lower for R and P. This is interesting and also applies to Figure 8, so it would be good to see Figure 8 split up by E/R/P, too.

5. The word “Figure” is missing before figure references.

6. Please proof-read for typography, punctuation and grammar.

---

> ### Author Response · Authors · 2018-11-26
> **Point-by-point address to reviewer's concerns [Part 1]**
>
> Thank you for the valuable feedback and comments. Below we address your comments point by point.
>
> The paper contains intriguing ideas about the benefits of sparse and predictive coding, and the direct comparison to biological data potentially broadens the impact of the work. However, major claims are unsubstantiated, and accuracy and clarity need to be improved to make the manuscript acceptable.
>
> Major concerns:
> 1. The authors claim that their architecture is more efficiency because it uses sparse coding of residuals. Implementation details and some quantitative arguments, ideally benchmarks, need to be provided to show that their architecture is actually more efficient than PredRNN++ and PredNet.
>
> Re: We might have given the wrong impression in our statements that our model was more efficient or faster than PredRNN++ and PredNet.  In fact, our training time is actually longer. We have now provided a Figure 4 (d) showing the training time for different versions of our model and PredRNN++ and PredNet. PredNet is fastest to train, because it is learning representations of sparse prediction residuals. Our B-B version of the network, which performed better than PredRNN++, took 10% longer to train than PredRNN++. This is not surprising because our model uses more loops than PredRNN++ and we used spatiotemporal blocks as data units and 3D convolutional LSTM for prediction, so naturally our model is more complex than PredRNN++ and will take longer time to train. Adopting Pan’s sparsificaiton scheme decreased B-B network's training time by 13% (comparing B-B (sparse) versus B-B (non-sparse) in Figure 4(d). Thus, the statement that sparse convolution improves efficiency is true. We have now added a benchmark comparison in Figure 4 (d) showing the training time and clarifying the limited contribution of sparse convolution to our model. Thank you for pointing out this potential confusion.
>
> 2. It is unclear whether the PredRNN++ should be compared to the C-C or C-F version of the network. Does the PredRNN++ have access to as many current and future frames as the C-C net? Is this a fair comparison? Please provide a clearer description of the different versions of your network and how they relate to the baseline models. That section in particular has many confusing typos (frame-by-block, block-by-frame abbreviations mixed up).
>
> Re:  Note, we have changed C (chunk) to B (block) in order to have more consistent notations and terminology. Is it a fair comparison with the baseline model PredRNN++? During testing, all the five networks (B-B, B-F, F-F, PredNet, PredRNN++) had access to the same number of frames (the first 20 frames) and have to predict the future 20 frames of the 40-frame test sets. During training, they were all trained on 40 frames movies of the training sets drawn from the same database. The comparison is fair in the sense that they have equal access to the same amount of information and they have to solve the same problem.  Both PredNet and PredRNN++ took in one frame at a time to predict one frame at a time, while our B-B took in a block of frames to predict a block of frames. PredRNN++ used a stack of LSTM to remember sequences and learn the feature transformation in the fashion of an autoencoder, while HPNet used the idea of a spatiotemporal block as well as a hierarchy of LSTM to do the same. Absolute fair comparison is difficult but we are fair at least in the amount of information available to each model, as reviewer asked.

---

> > ### Author Response · Authors · 2018-11-26
> > **Point-by-point address to reviewer's concerns [Part 2]**
> >
> > 3. In Figure 6, the authors claim that more layers lead to “better” representations. What does “better” mean? It is implied that the networks with more layers actually make the different motions more discriminable. Please quantify this. For example, a linear classifier could be trained on the neural activations. Also, how is this related to the rest of the paper? Do the authors claim that this result is unique to the proposed architecture? In that case, please provide a quantitative comparison to the PredNet or PredRNN++.
> >
> > Re: One of the key results (insights) of this work is that having more layers/modules leads to more semantically meaningful representations in the earlier layers/modules. Our t-SNE graph shows better clustering in module 1 for the six movements visually. We took reviewer’s excellent advice, and performed movement decoding using a linear classifier based on the representation of module 1 to quantify our intuition. Indeed, we found that the representation of the first module shows a significant decoding accuracy improvement in the classification of the six classes of movement patterns from chance (16%) to 26%.  We also found that the representation of module 4 of the 4-module network supports 63% decoding accuracy even though the network was never trained to discriminate the 6 movements using supervised learning with labelled data. We discussed this in the paper and provided more detailed results in the Appendix B.  For comparison with PredRNN++, we applied the same decoding to the output of their LSTM of each level, documented their results in Appendix B. Interestingly, the highest decoding results for  PredRNN++ is only 0.23 based on the LSTM output at layer 2, whereas HPNet’s highest decoding result is 0.63 based on module 4’s representation.  This shows that PredRNN’s hierarchy of LSTMs only has limited semantic clustering of the global movement patterns in its hierarchy partly because the real feature hierarchy might only be 2 layers in a 4-layer network.
> >
> > 4.In Figure 9, the presentation is highly confusing. Plots (c) to (h) are clearly made to look like the monkey data in (b) (nonlinear x-axes?), but show totally different timescales (training epochs vs. milliseconds). Please explain why it makes sense to compare these timescales. Also, what does it mean for a training epoch to have a negative value?
> >
> > Re: Yes. That is a terrible mistake. The x-axes should be time after stimulus onset. It has nothing to do with training epochs. We have made the correction.
> >
> > Minor comments:
> > 1.I don’t understand the “tension” between hierarchical feature representations and residual representations brought up in Section 2. Do the PredNet and PredRNN++ not contain a hierarchy of representations?
> >
> > Re:  PredNet has a hierarchy of representation for making predictions on prediction errors. That is, in PreNet, each layer’s LSTM is trying to predict the prediction errors observed in the earlier layer. PredRNN++ likely have a hierarchical representation of spatiotemporal features in the intermediate layers but remember that their highest layer output the prediction at the image level, so it is functions like an LSTM-based autoencoder. Our hierarchical prediction network (HPNet) is designed to address these conceptual deficiencies or problems (in terms of neural plausibility in our mind) in these two models by having both feedforward analyzed feature representation and feedback expected feature representation at every layer, and then compute prediction errors at each layer. It is a very simple conceptual framework common to many the classical hierarchical cortical processing model frameworks (Mumford, Ullman etc.). We now expand our Related Work section to provide a broader view on these issues.

---

> > > ### Author Response · Authors · 2018-11-26
> > > **Point-by-point address to reviewer's concerns [Part 3]**
> > >
> > > 2. Figure 1 is not fully annotated and could be clearer. What does the asterisk mean? Why are there multiple arrows between the P’s? What do the small arrows next to the big arrows mean? Please expand the legend. Consider using colors to differentiate between components.
> > >
> > > Re: Thank you for your advice. We have taken your suggestions to heart and revised Figure 1, and added more annotations and captions to make it more understandable.
> > >
> > > 3. I don’t understand Figure 4c. According to the text, this plot shows “effectiveness as a function of time”, but the x-axis is labeled “Layer Number”. What does “effectiveness over time” mean? What does the y-label mean (SSIM per day?)? What is “trunk prediction” (not mentioned anywhere in the text)?
> > >
> > > Re:  Sorry for the lack of clarity. The purpose of Figure 4© is to compare the effectiveness of B-B and B-F as a function of the number of layers (modules) utilized in the network. In the original Figure 4(c), where effectiveness is the ratio between SSIM performance and the amount of training time required, all we are trying to show is that the 4-module B-B network might be the optimal. We modified Figure 4 (c) to simply show SSIM as a function of the number of layers (modules), labeling SSIM points with the amount of training time required, so that readers can understand why we choose the 4-module network.
> > >
> > > 4. For Figure 9, it is pointed out that activity is expected to be lower for E neurons, but is also lower for R and P. This is interesting and also applies to Figure 8, so it would be good to see Figure 8 split up by E/R/P, too.
> > >
> > > Re:  We have decided to move the old Figure 8 to the Appendix to make more room to discuss our novel neurophysiological findings on prediction suppression effect in V1 and V2.  Upon the reviewer’s request, we now included in Appendix the responses of all three types of cells in HPNet for Meyer and Olson’s (2011) prediction suppression experiment.
> > >
> > > 5.The word “Figure” is missing before figure references.
> > >
> > > Re: Thank you, we have corrected that.
> > >
> > > The paper contains intriguing ideas about the benefits of sparse and predictive coding, and the direct comparison to biological data potentially broadens the impact of the work. However, major claims are unsubstantiated, and accuracy and clarity need to be improved to make the manuscript acceptable.
> > > The benefit of sparse convolution and residual coding of video has been demonstrated by Pan et al. (2018) in the context of video processing. It is also demonstrated in Lotter and Cox’s PredNet, though they might not have realized at the time that their predictive coding scheme actually has the benefit of learning sparse convolution kernel and has the benefit of computational efficiency. In the theoretical neuroscience community, sparse coding is considered mostly for coding efficiency, not for making computation efficient as well. We made this observation based on Pan et al’s contribution, and based on comparisons between our frame-to-frame model with Lotter and Cox’s PredNet, and our Block-to-Block model with and without Pan’s sparsification. We have documented all these in Figure 4C to clarify these issues, but this observation, though interesting, is really not the main contribution of the paper.

---

> ### Author Response · Authors · 2018-12-12
> **Response to reviewer 1's new comments on rebuttal [Part 1]**
>
> Reviewer 1’s comment on rebuttal: Disincentivize rushed work
>
> Comment: The authors addressed most of my comments. However, I think it needs to be taken into account how unfinished the original submission was. The authors admit to submitting the paper at the last minute, presumably hoping to use the review period to finish the paper. I know that this is common practice, but that does not make it acceptable. Submitting unfinished work with the hope to finish it later is unfair towards authors who submit finished papers in time. Allowing this behavior also exacerbates incentives to work fast rather than thoroughly. This leads to poor science. Finally, submitting unfinished work wastes reviewer time, eroding reviewer motivation to perform thorough initial reviews.
>
> The review process exists to encourage thorough work and good scientific practice.  In my  opinion, this  includes disincentivizing the last-minute submission of unfinished work.
>
> Authors’ responses:  We are sorry that reviewer 1 declined to re-assess our paper in its current form on the ground that (1) our paper was “unfinished” in the original submission, and (2) it would be unfair to our competitors who submitted “finished” papers in the original deadline. In our defense, our paper was “unfinished” only in terms of our presentation and English writing, missing typographical and a grammar check in certain sections of the paper. Our technical work and all the core results and graphs were finished by the time of submission and they have remained the same in the revision. We have added one subfigure(4D) and two Supplementary sections to satisfy Reviewer 1’s suggestions but these are not central to our paper. These additions are:  (1) Figure 4D was added to address Reviewer 1’s request that we make explicit the training time-performance comparison between our model and PredRNN and PredNet. We should further point out that training time was not a core contribution of our paper, nor is efficiency our major claim,  and Figure 4D thus is not critical.   (2) Supplementary  Section B was added to compare representations and decoding results between the representations of our models and the other models, also thanks to Reviewer 1’s suggestion.  We should point out that other papers on the same topic mostly reported and compared performance without providing insights and information on the representations – this includes PredNet and PredRNN.  (3) Supplementary Figure 11 was added also to satisfy reviewer 1’s suggestion on the Meyer and Olson’s experiment, and again that is not critical to the paper.  Because of that,  we moved the Meyer and Olson’s IT experiment to the Supplementary information.
>
> Thus, we argue that our technical work in fact was complete and finished by the time of the original submission. We have not changed our models, and we have not added significant new core result figures in the new revision.  If technical and conceptual contributions and experimental results are key for judging one paper against another, we believe our revision is NOT unfair to other competitors.
>
> We do agree and admit that our original submission’s writing is far from perfect and some of our presentation left much to be desired. We also are thankful for the reviewers’ helpful comments, and appreciated ICLR’s current policy of allowing revision of the manuscript for clarity of presentation and to address the reviewers’ questions and concerns. We have not tried to game the system as we have not upgraded our models or changed or upgraded our basic core findings and results. We did use this opportunity to polish the presentation of the paper based on reviewers’ suggestions and criticisms.

---

> > ### Author Response · Authors · 2018-12-12
> > **Response to reviewer 1's new comments on rebuttal [Part 2]**
> >
> > We tried our best to submit the best version of the paper by the deadline, but we did have our limitations. We were blind to our own imperfection at the time. We did not submit the paper by the deadline with the intent of “finishing the paper during the rebuttal period”. We appreciated the reviewers’ suggestions and careful reviews, and we did take the opportunities, as we believe to be permitted by ICLR policy, to improve our presentation.
> >
> > Indeed, we believe we have already suffered for the imperfection in our original presentation – Reviewer 2 for example maintained the original 7 score, even though (s)he appreciated the contributions of the paper even in the first round, and has clearly taken this issue into account;  Reviewer 3 increased to score to 7 from 3, as (s)he promised to do if we improved our writing, because (s)he could see the contribution and potential impact of the paper even in the original submission, despite our shortcomings in presentation in the first round. Had we done a better job in presentation and paper writing, we believe we would have been given even better scores as the paper would add diversity and significant values to ICLR contributions and would build connections between machine learning and neuroscience.
> >
> > While we can understand and sympathize with Reviewer 1’s philosophy, we also believe ICLR policy should be uniformly applied to all submissions on what kinds of revisions are allowed, and what constitutes the key aspects of the papers for comparison, judging and final evaluation for acceptance.

---

### Official Review · AnonReviewer2 · 2018-11-03
**Interesting bio-inspired sequence prediction network explaining familiarity effects in early and late visual system.**

**Rating:** 7
**Confidence:** 3

**Review:**

The authors propose a biologically inspired ANN to predict a video sequence, that performs better than previous biologically inspired video sequence predictors (>PredNet and >PredRNN+).  Their model also accounts for familiarity effects (i.e. decrease in neural activations when repeatedly presenting the same visual sequence) found in primate early visual system V1/V2 (data recorded for this article) and late visual system IT.

This work is interesting because it proposes a sequence prediction technique that accounts well for familiarity effects found in different regions of the visual system.

However one of the claims does not seem supported by data:

1. The authors claim repeatedly that using the prediction error framework is computationally more efficient than alternatives but they do not show this.

Furthermore, the article would benefit from the following clarifications:

2. It is unclear how their network performance compares to state-of-the-art NON neurally plausible models of sequence prediction.

3. It is unclear from the introduction how they modified the network proposed by (Pan et al) to obtain their network.

4. "The SSIM index over time shows that the C-C method is more effective than C-F method, for C-F method performs better than C-C method in the short term perdiction when ground truth images are provided, but setting sliding window is too time-consuming, much more than the performance increase"
Please clarify this statement.

5. Macaque experiments: Some experiments on macaques were performed for this article, but there is no mention of ethical guidelines and whether they were respected.

6. Many typos are present in the text!

I believe this work at the intersection of deep learning and neuroscience is an interesting contribution for both fields. However, the paper would benefit from these clarifications and a thorough proof-reading for the many typos present in the text.

---

> ### Author Response · Authors · 2018-11-26
> **Point-by-point address to reviewer's concerns [Part 1]**
>
> Thank you for the valuable feedback and comments. Below we address your comments point by point.
>
> 1. The authors claim repeatedly that using the prediction error framework is computationally more efficient than alternatives but they do not show this.
>
> Re: In Pan et al.’s CVPR 2018 paper, Recurrent Residual Module for Fast Inference in Videos, they have shown the efficiency of using residual error sparse convolution over normal convolution neural network for video processing. We also found that PredNet runs much faster than our F-F model because it builds a hierarchy of prediction errors, i.e. the errors of errors, which should be sparse and hence it also trains much faster than PredRNN++, which presumably learn some hierarchy of spatiotemporal memories. Our model HPN took more time even than PredRNN++ (by 10%) because it processed video in the unit of spatiotemporal blocks with spatiotemporal convolution rather than frame by frame as in the other two baseline models. We now added a quantitative comparison of training time of the different models in Figure 4(d). Using Pan et al’s sparse convolution scheme in our feedforward path saved our training time by about 15%.
>
> 2. It is unclear how their network performance compares to state-of-the-art NON neurally plausible models of sequence prediction.
>
> Re: PredNet a neurally inspired model. PredRNN++ is the state-of-the-art NON neurally plausible model for video prediction. In PredRNN++, which is a paper published in ICLM 2018, the authors documented its performance against other computer vision models for video sequence prediction of the same kind, and showed that it is the state-of-the-art performing model for this task. Thus, we thought it sufficient to compare our model against PredRNN++.
>
> 3. It is unclear from the introduction how they modified the network proposed by (Pan et al) to obtain their network.
>
> Re: Pan et al. network learns a dense convolutional kernel to process the first frame, but learns a sparse convolution kernel to process the subsequent frames. We just learn one sparse kernel for all the frames, including the first to reduce the parameters by half. We feel this parsimonious approach, even though it is slightly less accurate in the beginning prediction, is more reasonable and neurally plausible. We discuss this in the paper. Overall, the adoption of Pan’s idea reduced our training time by about 13-15% and is not a critical part of the current model, although sparse convolution might be an important design principle in future refinements of the networks. We now separate the main idea having a DCNN feedforward path from this minor refinement into Figure 1(a) and Figure 1(b) to facilitate conceptual understanding.
>
> 4. "The SSIM index over time shows that the C-C method is more effective than C-F method, for C-F method performs better than C-C method in the short term prediction when ground truth images are provided, but setting sliding window is too time-consuming, much more than the performance increase" Please clarify this statement.
>
> Re: We apologized for our lack of clarity in this explanation. Now we changed Chunk to Block for a more accurate and consistent exposition. We have rewritten those explanations and hope they are clear now. Essentially, the B-F (used to be called C-F) method takes in a spatiotemporal block as input to predict an individual frame. In this method, we have to move essentially frame by frame to predict one frame at a time, using a block of frames as input.  The B-B (used to be called C-C) method can take a temporal stride as large as 5 frames at a time (if the spatiotemporal block contains 5 frames). B-F can be considered as B-B with sliding window of 1 frame. Obviously, the B-F method produces a more accurate near-term prediction than B-B, or F-F, but it is time consuming and underperforms overall. B-B however is faster, when it takes a stride of 5 frames, and actually produces more accurate results for long range predictions.
>
> 5. Macaque experiments: Some experiments on macaques were performed for this article, but there is no mention of ethical guidelines and whether they were respected.
>
> Re: Thank you for reminding us. We have now added a footnote in the description of the experiment stating that “All experimental procedures were approved by the XX University Institutional Animal Care and Use Committee and were in compliance with the guidelines set forth in the United States Public Health Service Guide for the Care and Use of Laboratory Animals.”
>
> 6. Many typos are present in the text!
>
> Re:  Yes, our apologies. We have revised our paper very carefully and extensively.

---

> > ### Author Response · Authors · 2018-11-26
> > **Point-by-point address to reviewer's concerns [Part 2]**
> >
> > 7. This work is interesting because it proposes a sequence prediction technique that accounts well for familiarity effects found in different regions of the visual system. … I believe this work at the intersection of deep learning and neuroscience is an interesting contribution for both fields. However, the paper would benefit from these clarifications and a thorough proof-reading for the many typos present in the text.
> >
> > Re.  Thank you for your recognition and appreciation of the contributions of our work. We have revised our paper very carefully and tried to better explain why this is indeed an interesting (and important) contribution to both fields.

---

### Official Review · AnonReviewer3 · 2018-11-05
**SOTA results in video prediction and interesting analysis but the presentation is severely lacking clarity**

**Rating:** 7
**Confidence:** 3

**Review:**

Summary:
The paper presents a novel architecture for video prediction consisting of a feed-forward path with sparse convolutions and an LSTM generating predictions of chunks of video based on the sequence of input chunks. A feedback path links the LSTMs of the different sparse prediction modules. Experiments in video prediction are performed on moving-MNIST and the KTH action recognition dataset and the model achieves state-of-the-art performance on both. Interestingly, the model is exhibits prediction suppressions effects as have been observed during neurophysiological experiments in the inferotemporal cortex of macaque monkeys. The proposed method exhibits prediction suppression effects also in the lower layers, motivating a neurophysiological experiment in the earlier V1/V2 regions, which yielded an observation similar to the model’s prediction.

Strengths:
The performance improvements over competing methods on Moving-MNIST and KTH presented in the experimental section are significant. The analysis seems fairly thorough.

Weaknesses and requests for clarification:
- The description of the sparse predictive module is difficult to follow, and I am not sure I understood it completely. I find it a bit unintuitive to start the description with the errors, instead of explaining what is computed from beginning to end. The section reads more like a loose description of isolated parts instead of an integrated whole. Maybe walking the reader step-by-step through one complete iteration of the computation helps to clarify this. Also, not every character in equations 1-5 and the algorithm has been defined. For example, what is L?
- The text makes it sound like the idea of using 3d convolutions in a convLSTM is novel. 3D convLSTMs have been previously used in 3d vision, see
Choy, C. B., Xu, D., Gwak, J., Chen, K., & Savarese, S. (2016, October). 3d-r2n2: A unified approach for single and multi-view 3d object reconstruction. In European conference on computer vision (pp. 628-644). Springer, Cham.
The application of 3d convLSTMs to video might be new, but the mentioned paper by Choy et al. (2016) should be cited.
- You mention that padding is used for rows and columns. Are you using padding on the temporal axis as well?
- The paper seems to be written in a rush, as it contains way too many typos and grammar mistakes, e.g. “a hierarchical of” (should be “a hierarchy of“ or just “hierarchical”), “feedforwad”, “Expriment” (section 4 heading), “achievedbetter”, “trained monkeys to image pairs”, “pervious”, “perserves”, “processure”, “sequnence” “viusal”. Many typos could have been caught by a spellcheck! This would improve readability a lot!
- The citations are not properly formatted: (1) If the author names are used as part of the sentence, use e.g. Lotter et al. (2016), else (2) If the author names are not part of the sentence, use (Lotter et al., 2016). These two styles are mixed randomly in the current draft. This makes the manuscript, which already contains a lot of language mistakes, difficult to read.
- Abbreviations that are used but not introduced: CNN, IT, PSTH, DCNN, LSTM.
- The related work section could benefit from referring to some of the related work in neuroscience.
- Adding a sentence explaining the intuition behind using SatLU in equation (1) might be helpful

To summarize my feedback: I think experimental results and analysis are strong, but the presentation is strongly lacking! The description of the approach definitely needs to be improved to make replication of the results easier. It might help to have someone who doesn’t know the model already read the description and explain it back to you while revising the draft. I hope I could provide some helpful suggestions. I would recommend the manuscript for acceptance, if the presentation is significantly improved!

---

> ### Author Response · Authors · 2018-11-26
> **Point-by-point address to reviewers’ concerns [Part 1]**
>
> Thank you for the valuable feedback and comments. Below we address your comments point by point.
>
> 1. The description of the sparse predictive module is difficult to follow, and I am not sure I understood it completely. I find it a bit unintuitive to start the description with the errors, instead of explaining what is computed from beginning to end. The section reads more like a loose description of isolated parts instead of an integrated whole. Maybe walking the reader step-by-step through one complete iteration of the computation helps to clarify this. Also, not every character in equations 1-5 and the algorithm has been defined. For example, what is L?
>
> Re: Now we call each layer a ‘Cortical Module (CM)’ and provide a more concise and precise description of the model and algorithm in section 3. We also provide a step-by-step description of the flow of the algorithm per your advice. For clarity, we decompose the description of the feedforward path into Figure 1(a) and Figure 1(b) into a normal DCNN path and a sparsified DCNN part to make it more understandable. The feedforward path is just a normal convolutional neural network but it is trainable by self-supervised learning because its feedforward input does project to the LSTM in each layer. The sparse convolution scheme (Figure 1b) only serves to make it more efficient (see Figure 4d), and is not really a critical part of the model.
>
> 2. The text makes it sound like the idea of using 3d convolutions in a convLSTM is novel. 3D convLSTMs have been previously used in 3d vision, see Choy, C. B., Xu, D., Gwak, J., Chen, K., & Savarese, S. (2016, October). 3d-r2n2: A unified approach for single and multi-view 3d object reconstruction. In European conference on computer vision (pp. 628-644). Springer, Cham. The application of 3d convLSTMs to video might be new, but the mentioned paper by Choy et al. (2016) should be cited.
>
> Re: Thank you for pointing that out. We added the citation of Choy et al. We believe, as you pointed out, that using 3D convLSTM in video, especially for video prediction, might be new though it seems to be an obvious thing to do. But we don’t think this is the main contribution of the paper.
>
> 3. You mention that padding is used for rows and columns. Are you using padding on the temporal axis as well?
>
> Re: We used padding on both spatial and temporal domains.
>
> 4. The paper seems to be written in a rush, as it contains way too many typos and grammar mistakes, e.g. “a hierarchical of” (should be “a hierarchy of“ or just “hierarchical”), “feedforwad”, “Experiment” (section 4 heading), “achieved better”, “trained monkeys to image pairs”, “pervious”, “perserves”, “processure”, “sequnence” “viusal”. Many typos could have been caught by a spellcheck! This would improve readability a lot!
>
> Re: Yes, absolutely. We are embarrassed and are terribly sorry. Indeed, the paper was written in a rush. We submitted the paper literally in the last minute, pushing the submit button 1 minute before the deadline. Well, that is obviously not a good excuse, and we are very grateful indeed that the reviewers are still willing to spend the time to read the paper despite its obvious shortcomings! We hope we have redeemed ourselves by putting in an enormous amount of effort into this revision.
>
> 5. The citations are not properly formatted: (1) If the author names are used as part of the sentence, use e.g. Lotter et al. (2016), else (2) If the author names are not part of the sentence, use (Lotter et al., 2016). These two styles are mixed randomly in the current draft. This makes the manuscript, which already contains a lot of language mistakes, difficult to read.
>
> Re:  Yes. We agreed and corrected them accordingly.
>
> 6. Abbreviations that are used but not introduced: CNN, IT, PSTH, DCNN, LSTM.#
>
> Re: Our bad. Now, we added the full names of each abbreviated term before using them and tried to minimizes the use of special terminologies by calling IT inferotemporal cortex and PSTH temporal responses of the neurons.
>
> 7. The related work section could benefit from referring to some of the related work in neuroscience.
>
> Re: We have added more background from theoretical neuroscience -- Mumford’s ideas on analysis by synthesis and Ullman’s counter-stream model, which is the inspiration of the development of our model. We also provided some recent neurophysiological studies on prediction errors in the inferotemporal cortex (Meyer and Olson 2012), as well as prediction related memory recall phenomena in the primary visual cortex (V1) of mice (Han et al. 2008, Xu et al. 2012). Our study on V1 and V2 neurons’ sensitivity to memory of familiar complex video episodes is novel. We moved our simulation results of Meyer and Olson (2012) to the Appendix to yield room for some additional clarifying discussion on this experiment.

---

> > ### Author Response · Authors · 2018-11-26
> > **Point-by-point address to reviewers’ concerns [Part 2]**
> >
> > 8. Adding a sentence explaining the intuition behind using SatLU in equation (1) might be helpful.
> >
> > Re: SATLU is a saturating non-linearity set at the maximum pixel value: SatLU(x; p_{max}):= min(p_{max}, x). Definitions: f is non-saturating iff (|limz→−∞f(z)|=+∞)∨|limz→+∞f(z)|=+∞), f is saturating iff ff is not non-saturating, as we now explained it more clearly in Section 3.4.
> >
> > 9. Strengths: The performance improvements over competing methods on Moving-MNIST and KTH presented in the experimental section are significant. The analysis seems fairly thorough.
> >
> > Yes, our analysis is not perfect, but better than many other state-of-the-art video prediction models which did not provide representational analysis to reveal the underlying reasons explaining why their models actually work better.
> >
> > 10. To summarize my feedback: I think experimental results and analysis are strong, but the presentation is strongly lacking! The description of the approach definitely needs to be improved to make replication of the results easier. It might help to have someone who doesn’t know the model already read the description and explain it back to you while revising the draft. I hope I could provide some helpful suggestions. I would recommend the manuscript for acceptance, if the presentation is significantly improved!
> >
> > Yes. Thank you very much for all the helpful suggestions and generosity despite our shortcomings.  We hope our serious revision of our manuscript will allow it to meet the standard of excellence for ICLR.

---

> > > ### Comment · AnonReviewer3 · 2018-12-09
> > > **My concerns have been addressed!**
> > >
> > > I believe all points I raised were addressed in the revision. I increased the score, recommending the paper for acceptance.

---

> > > > ### Author Response · Authors · 2018-12-09
> > > > **Thank you**
> > > >
> > > > Thank you for taking the time to reevaluate our paper. We really appreciate it.

---

### Author Response · Authors · 2018-11-26
**General responses to the reviewers and the program committee**

We thank the reviewers for reading our paper carefully, despite our poor presentation and numerous typographical mistakes. We thank reviewer 1 for recognizing that “the experimental results and analysis are strong” and for stating that our paper would be acceptable if the presentation were improved. We also thank reviewer 2 for recognizing that our “work is interesting because it proposes a sequence prediction technique that accounts well for familiarity effects found in different regions of the visual system.” and that this work is “at the intersection of deep learning and neuroscience is an interesting contribution for both fields.” Finally, we thank reviewer 3 for recognizing “the paper contains intriguing ideas about the benefits of sparse and predictive coding, and the direct comparison to biological data potentially broadens the impact of the work”.
We have seriously revised and proofread the paper and we hope that the current version will receive a more favorable score.

Here is a highlight of the contribution of our paper:
(1) Our model HPNet provides an alternative to the Predictive Coding model, the basis of PredNet,  which is quite popular in neuroscience. The model might be the first computational competent hierarchical neural cortical model that implements the classical computational framework for cortical processing (analysis by synthesis, counter-stream architecture, interactive activation and adaptive resonance) and is competitive in solving real computer vision problems. It works better because the synthesis is no longer done by simple deconvolution or multiplying through feedback connection weights as in classical models but are generated by the gated recurrent circuits in LSTM. It shows that feature hierarchy works better than prediction error hierarchy (PredNet).
(2) We provided thorough analysis of the representations of our network to understand what could be the reasons that the network is working better than PredNet or PredRNN++.  We discovered that recurrent feedback has reshaped the representations of the early modules (layers), making “neurons” in the bottom modules sensitive to memories of global movement patterns, i.e. more abstract concepts, rather than just local spatiotemporal features in their receptive fields. The semantic clustering of global movement patterns might have contributed to better long-range video prediction by facilitating the relationship learning of movement patterns. Thanks to the reviewer’s suggestion, we performed a decoding experiment and showed quantitatively that global movement patterns have indeed become more segregated and discriminable in the representation of the early modules due to feedback, and more importantly, HPNet’s hierarchical representations contain semantic clusters, achieving 63% decoding accuracy in the 4th layer for classifying movement patterns, while PredRNN’s and particularly PredNet’s hierarchical LSTMs provide little semantic information (all layers) for decoding the global movement patterns (< 26%) – See Appendix B.
decoding results (for all layers) are either close to chance or
(3) The most interesting part of our story is that we found that neurons in the early visual cortex of awake monkeys developed similar sensitivity to memories of global movement patterns in video when they are repeatedly exposed to a set of movies. This discovery, under the ``computational illumination’’ of HPNet, provides new insights and concrete evidence to the potential computational logic of recurrent feedback in the cortex, and gives us more faith on the neural plausibility of t this class of predictive self-supervised learning models.

We believe these core claims are substantiated by our experiments, analysis and data. The idea that sparse coding might make convolution fast (reviewer 3) is not really our contribution. Pan et al. (2018) have provided experimental results on video processing that show sparsifying the representation can speed up computation. (see our point-by-point response to reviewer 3 for details). However, we do apologize if we inadvertently made statements that gave the mistaken impression that HPNet is faster to train than PredNet and PredRNN++. We have added a graph (Figure 4d) documenting the training time and performance of the different models, which clearly shows PredNet is the fastest and ours is the slowest to train. HPNet is processing spatiotemporal blocks with 3D convolutional LSTM, and it has a feature hierarchy in both its feedforward and feedback paths absent in the other two models, so naturally it would take longer to train than PredRNN++. It is only with sparsification and taking longer strides in the sliding window that we can train HPNet  at comparable times.

We hope the reviewers and the program committee seriously consider our revised paper for its potential impact in both neuroscience and machine learning.

---

### Meta-Review · Area_Chair1 · 2018-12-14
**Significant revisions in review**

**Confidence:** 4
**Recommendation:** Reject

**Metareview:**

There was major disagreement between reviewers on this paper. Two reviewers recommend acceptance, and one firm rejection. The initial version of the manuscript was of poor quality in terms of exposition, as noted by all reviewers. However, the authors responded carefully and thoroughly to reviewer comments, and major clarity and technical issues were resolved by all authors.

I ask PCs to note that the paper, as originally submitted, was not fit for acceptance, and reviewers noted major changes during the review process. I do believe this behavior should be discouraged, since it effectively requires reviewers to examine the paper twice. Regardless, the final overall score of the paper does not meet the bar for acceptance into ICLR.